# Probing Majorana localization of a phase-controlled three-site Kitaev chain with an additional quantum dot

Alberto Bordin ®[1,4], Florian J. Bennebroek Evertsz'[1,4], Bart Roovers ®[1,4], Juan D. Torres Luna[1,4], Wietze D. Huisman ®[1], Francesco Zatelli ®[1], Grzegorz P. Mazur ®[1], Sebastiaan L. D. ten Haaf ®[1], Ghada Badawy[2], Erik P. A. M. Bakkers ®[2], Chun-Xiao Liu ®[1], Rubén Seoane Souto[3], Nick van Loo ®[1] & Leo P. Kouwenhoven ®[1] ✉

Few-site implementations of the Kitaev chain offer a minimal platform to study the emergence and stability of Majorana bound states. Here, we realize two- and three-site chains in semiconducting quantum dots coupled via superconductors, and tune them to the sweet spot where zero-energy Majorana modes appear at the chain ends. We demonstrate control of the superconducting phase through both magnetic field and sweet-spot selection, and fully characterize the excitation spectrum under local and global perturbations. All spectral features are identified using the ideal Kitaev chain model. To assess Majorana localization, we couple the system to an additional quantum dot. The absence of energy splitting at the sweet spot is compatible with high-quality Majorana modes, despite the modest chain size.

The Kitaev chain model is one of the simplest implementations of topology in condensed matter[1]. It describes a spinless chain of $N$ fermionic sites ($c_n$) with energies $\mu_n$, nearest-neighbor hoppings $t_n$, and superconducting-like pairings $\Delta_n$[2] (Fig. 1a, b):

$$H_N = \sum_{n=1}^{N} \mu_n c_n^\dagger c_n + \sum_{n=1}^{N-1}(t_n c_n^\dagger c_{n+1} + \Delta_n c_n^\dagger c_{n+1}^\dagger + h.c.) \tag{1}$$

For $N \to \infty$, Kitaev showed that the chain hosts two Majorana bound states (MBSs), one at each end of the chain, which are exponentially close to zero energy and topologically protected - meaning that no local perturbation of the Hamiltonian can couple them[1]. This makes the Kitaev chain a promising candidate for a robust quantum memory, as the dephasing time would be inversely proportional to the energy splitting of the MBSs[3].

Remarkably, even minimal Kitaev chains of just two sites can host unpaired MBSs, which are exactly at zero energy if $|t_1| = |\Delta_1|$ and $\mu_1 = \mu_2 = 0$[4]. However, they are not topologically protected as their energy splits linearly with $|t_1| - |\Delta_1|$, motivating the label of *poor man's* Majoranas[4]. The transition from the poor man's ($N = 2$) to the topological regime ($N \to \infty$) is an active field of research[2,5–11]. The general trend is an increase in the protection from perturbations as the chain is scaled up[9], but the trajectory is not necessarily monotonic: in particular, three-site chains can potentially be worse than two-site ones due to next-nearest-neighbor hoppings[7,9,11] and four-site chains could be worse than three-site ones due to even-odd effects[6]. Recently, two- and three-site Kitaev chains were experimentally realized in hybrid semiconducting-superconducting nanowires[12,13] and two-dimensional electron gases[14,15]; attracting substantial experimental[16–21] and theoretical[22–35] attention to the understanding of the underlying physics.

In this work, we realize two- and three-site chains in a single device, describe their phenomenology, and test the Majorana quality by coupling them to an additional quantum dot. This has both a

[1]QuTech and Kavli Institute of NanoScience, Delft University of Technology, Delft, The Netherlands. [2]Department of Applied Physics, Eindhoven University of Technology, Eindhoven, The Netherlands. [3]Istituto de Ciencia de Materiales de Madrid (ICMM), Consejo Superior de Investigaciones Científicas (CSIC), Madrid, Spain. [4]These authors contributed equally: Alberto Bordin, Florian J. Bennebroek Evertsz', Bart Roovers, Juan D. Torres Luna. ✉e-mail: l.p.kouwenhoven@tudelft.nl

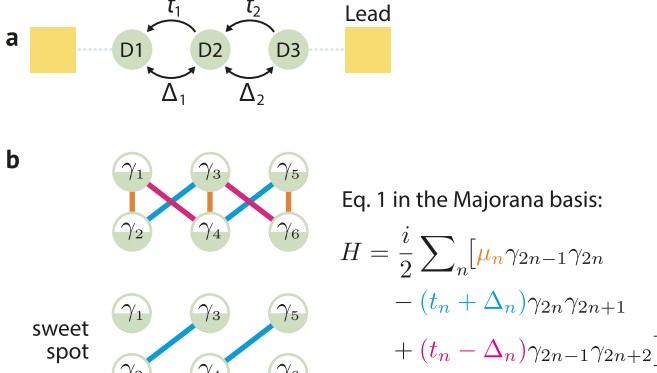

**Fig. 1 | Forming Kitaev chains in hybrid nanowires. a** Schematic of a three-site Kitaev chain coupled to two metallic leads (yellow). $t_n$ and $\Delta_n$ couple neighboring quantum dots. **b** When all inter-dot couplings are real numbers, a three-site chain can be represented in the Majorana basis with this simple diagram: $\mu_n$ couples the Majoranas within the same site, whereas $(t_n + \Delta_n)$ and $(t_n - \Delta_n)$ couple Majoranas of neighboring sites. At the $\mu_n = 0$, $t_n = \Delta_n$ sweet spot, there is one unpaired Majorana mode at each chain end. **c** A false-colored scanning electron micrograph of the

reported device, including circuit elements and gate voltage labels. An InSb nanowire (green) is deposited on an array of bottom gates (pink), which can define three quantum dots: D1, D2, and D3. They are coupled by two InSb-Al hybrids[16], which are connected by a superconducting loop (blue), grounded. There is also the possibility of defining an additional quantum dot (AD) on either side. Two Cr/Au contacts (yellow) can probe the local density of states on either side of the device via tunneling spectroscopy.

practical and a fundamental purpose. For technological applications, it is important to understand whether or not three-site Kitaev chains are better than two-site ones and whether it is beneficial to scale up the chain even further. For a fundamental understanding of the onset of topology, it is insightful to investigate how the partial protection from perturbations evolves in these finite-size Kitaev chains. In our previous work[13], we investigated the stability against internal perturbations of the Hamiltonian terms (1). Here, we test the stability against the simplest external addition to the Hamiltonian: one extra energy level, provided by an additional quantum dot[36,37]. Theory predicts that if the MBSs are well localized at the chain ends, then the additional quantum dot can couple only to a single Majorana mode and, therefore, nothing happens to its energy; if, however, there is a finite overlap between the MBS wavefunctions, so that the quantum dot can couple to both, then the MBSs gain a finite energy splitting[24,38,39].

## Results

### Realization of two- and three-site Kitaev chains

Our device consists of a semiconducting nanowire (InSb)[40] deposited on top of an array of bottom gates, separated by a dielectric layer (Fig. 1c). Two aluminum strips induce superconductivity in two nanowire sections, forming semiconducting-superconducting hybrids. They are connected in a loop geometry so that an out-of-plane magnetic field $B_z$ can tune the relative phase difference $\varphi$. Besides the aluminum, which is grounded, there are two additional contacts made of gold, which we use to probe the local density of states via tunneling spectroscopy[12]. They are connected to respective voltage sources ($V_L$, $V_R$), current meters ($I_L$, $I_R$), and standard lock-ins to measure the differential conductances ($g_L \equiv \frac{dI_L}{dV_L}$, $g_R \equiv \frac{dI_R}{dV_R}$). Further nanofabrication details are reported in Methods.

To form a Kitaev chain, we apply voltages to the bottom gates to define three quantum dots separated by the two hybrid sections[18]. A magnetic field $B_x = 175$ mT parallel to the nanowire ensures spin polarization of all quantum dots (verified in Fig. S7). The dot electrochemical potentials $\mu_n$ are controlled by the plunger gate voltages $V_{D1}$, $V_{D2}$, and $V_{D3}$, respectively, while the inter-dot couplings $t_n$ and $\Delta_n$ are tuned with the hybrid gate voltages $V_{H1}$ and $V_{H2}$[16,22]. There is also the option of forming an additional quantum dot (labeled AD) to test the Majorana localization. This additional quantum dot is used for Fig. 4; otherwise, there is a single tunneling barrier separating the left gold

contact from D1. All bottom gate settings are available in the linked repository[41].

To define two-site Kitaev chains, we use the following procedure. After forming the D1, D2, and D3 dots (their characterization is reported in Fig. S7), we set D3 off-resonance by adding ~ 5 mV to $V_{D3}$. A two-site chain comprising D1 and D2 is formed by balancing the $t_1$ and $\Delta_1$ couplings by fine-tuning $V_{H1}$[12,16]. The *poor man's Majorana sweet spot* $|t_1| = |\Delta_1|$ is reached when the D1-D2 charge stability diagram of Fig. 2a shows a cross shape. The spectrum at the center of the cross (corresponding to $\mu_1 = \mu_2 = 0$) is reported in Fig. 2b and shows a $|2t_1| = |2\Delta_1| \approx 30$ μeV energy gap between the zero-bias conductance peak (ZBP) and the first excited state (see Fig. S8 for a fit of the spectrum). Similarly, a two-site chain on the right of the device can be formed by setting D1 off-resonance and balancing the $t_2$ and $\Delta_2$ couplings between D2 and D3 by fine-tuning $V_{H2}$. At the sweet spot (Fig. 2c), we find a $|2t_2| = |2\Delta_2| \approx 20$ μeV energy gap (Fig. 2d). With these settings, the inter-dot couplings $t_n$ and $\Delta_n$ are much smaller than the Zeeman splitting ($E_Z \gtrsim 200$ μV, Fig. S7) and the minimum energy of the Andreev bound states (ABSs) located in the hybrid sections ($E_{ABS} \gtrsim 100$ μV, Fig. S17).

At this point, it is sufficient to bring D1 back on resonance to obtain a three-site chain with $\mu_n = 0$ and $|t_n| = |\Delta_n|$ for all $n$. We recall that the $t_n$ and $\Delta_n$ couplings are, in principle, complex numbers. Their phase is irrelevant in two-site chains, whereas in three-site chains a single non-trivial phase degree of freedom $\varphi$ remains[2,13]. This can be tuned with out-of-plane field $B_z$[15], as shown in Fig. 2e (the corresponding simulation is reported in Fig. S9j). We observe a periodic spectrum with a period $T_\varphi = 7.5$ mT. At $B_z = 0$ (mod $T_\varphi$), there is a maximum gap separating the ZBP from the first excited state, whereas they merge at $B_z = T_\varphi/2$ (mod $T_\varphi$). This means that the phase difference is $\approx 0$ at zero flux. This does not seem to be a coincidence: previous works predicted $t_n$ and $\Delta_n$ to be real when there is no magnetic field component perpendicular to a hybrid Rashba nanowire[2,32,42]. If all the $t_n$ and $\Delta_n$ are real, then $\varphi$ is either 0 or $\pi$. Indeed, we sometimes observe $\varphi \approx 0$ and sometimes $\varphi \approx \pi$ at $B_z = 0$. For instance, Fig. 2f shows an example of $\varphi \approx \pi$ at $B_z = 0$ measured in another three-site sweet spot using different QD and hybrid gate settings. Overall, we characterized the phase of 11 different sweet spots. They are reported in Figs. S6 and S18 and summarized in Fig. 2g: at $B_z = 0$, we observed $\varphi \approx 0$ seven times and $\varphi \approx \pi$ four times. This bimodal nature of the observed phase behavior has significant implications for scaling to long chains[32].

## Tuning neighbouring QD pairs to the PMM sweet-spot

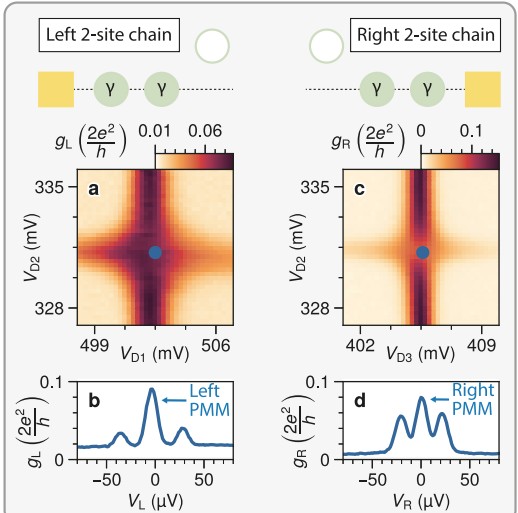

## Tuning the phase $\varphi$ in three-site Kitaev chains

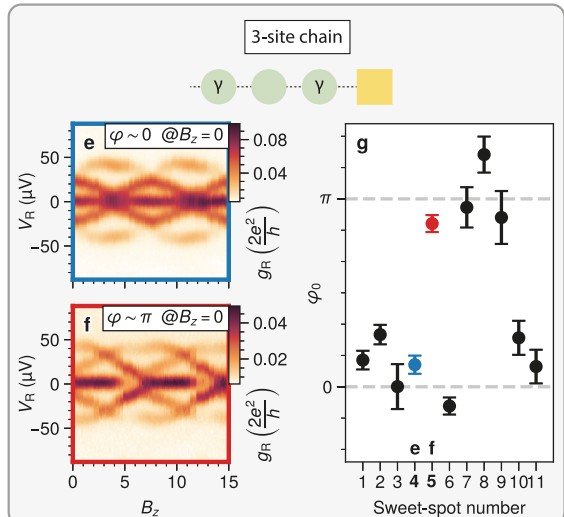

**Fig. 2 | Tuning process. a–d.** Two-site chains at the *Poor Man's Majorana* sweet spot. The left chain is measured while keeping D3 off-resonance, whereas the right chain is measured while keeping D1 off-resonance. Panels (**a**) and (**c**) show the D1-D2 and D2-D3 charge stability diagrams measured with standard lock-in techniques at zero DC voltage bias ($V_L = V_R = 0$). The blue dots mark the points where the finite-bias spectroscopy reported in panels (**b**) and (**d**) is measured. **e–g** Three-site chains. **e, f** Examples of tunneling spectroscopy as a function of the out-of-plane magnetic field $B_z$. Both panels show a periodic spectrum with a 7.5 mT period. However, opposite behavior is seen at $B_z = 0$: in panel (**e**), there is an energy gap, while in panel

(**f**) the gap is nearly closed. This suggests that at zero out-of-plane field, the phase is close to zero in panel (**e**) and close to $\pi$ in panel (**f**). The same analysis is repeated at multiple Kitaev chain sweet spots (see Fig. S18 for a full inventory) and summarized in panel **g**. Overall, at $B_z = 0$ mT we measured $\varphi \approx 0$ seven times and $\varphi \approx \pi$ four times. The blue and red markers correspond to panels (**e**) and (**f**), respectively. The phases shown in panel (**g**) are extracted by fitting the Kitaev chain Hamiltonian to the tunneling spectroscopy data; details of the fitting procedure are provided in Methods and in the Supplementary Information.

In three-site chains, tuning the phase is straightforward, as a single parameter, $B_z$, suffices. However, in longer chains, each additional site would typically require an extra phase-control parameter $\varphi_n$. Remarkably, Fig. 2g suggests that the phase difference can be set to $\approx 0$ at $B_z = 0$ by the sweet-spot choice, eliminating the need for additional tuning knobs: an arbitrarily long Kitaev chain can be fully tuned electrostatically and at zero out-of-plane field by selecting, in sequence, sweet spots that preserve $\varphi_n \approx 0$[32,43]. We note that in our experiment, $\varphi_n \approx 0/\pi$ at $B_z = 0$ with finite deviations: the field uncertainty is ~0.2 mT while the standard deviation from 0 or $T_\varphi/2$ is 0.5 mT (see also Fig. S6). This indicates a possible breaking of the complex conjugate symmetry, which restricts the phases to be real in a hybrid Rashba nanowire[42]. This symmetry breaking could stem from magnetic orbital effects on the QD or ABS orbitals or from an applied magnetic field not precisely perpendicular to the Rashba spin-orbit field[32,43]. Importantly, there is no need to tune to zero precisely for practical applications: it was calculated that, as long as $\varphi_n < \pi/2 \ \forall \ n$, the Kitaev chain has a finite topological gap[32]. All seven sweet-spots where we measured $\varphi_n \approx 0$ at $B_z = 0$ are within this bound.

In the rest of the manuscript, we focus on the sweet spot of Fig. 2e (number 4 in Fig. 2g) and set $B_z = 0$. $B_x$ is still at 175 mT to ensure that all QDs are polarized. The spectra as a function of various QD detunings are reported in Fig. 3. The first row shows tunneling spectroscopy from the left lead, while the second row shows spectroscopy from the right lead. Different columns show different types of QD detunings: in the first three columns, each dot is detuned separately, in the fourth column, two dots are detuned simultaneously, and in the fifth column, all dots are detuned together. As observed in our previous work[13], the ZBP persists for any local perturbation of one or even two QDs. Only the global perturbation of three QDs altogether is able to split the ground state degeneracy (Fig. 3i, j).

Turning our attention to the excited states, we highlight a feature that might go unnoticed: in the first and third columns (Fig. 3a, b, e, f), there is a single excited state per panel. This reflects the behavior of an

ideal three-site Kitaev model, illustrated in the schematics below. Every site is represented by two Majoranas that couple locally with amplitude $\mu_n$, while $t_n + \Delta_n$ couple neighboring sites and lead to the excited states marked in dark and light blue[15]. As long as $\mu_2 = 0$, the excited states are disconnected: a local probe on the leftmost site cannot sense the $t_2 + \Delta_2$ state (light blue), whereas a local probe on the rightmost site cannot sense the $t_1 + \Delta_1$ state (dark blue). This separation of the excited states breaks down as soon as $\mu_2 \neq 0$ and, indeed, we observe the appearance of a second excited state in each panel where $V_{D2}$ is detuned (c, d, g, h, i, and j). In all panels, the second excited state disappears as $\delta V_{D2}$ approaches 0 mV. We note that this disappearance of the second excited state can happen only at $\varphi = 0 \ (\text{mod} \ 2\pi)$. All spectra are reproduced by the numerical conductance simulations with $\varphi = 0$ reported in Fig. S9a–h.

Here, having the $t_1 + \Delta_1$ state visible only from the left and the $t_2 + \Delta_2$ state visible only from the right is direct evidence of the-localization of the excited states. Unfortunately, it does not prove the ground state localization as well. Therefore, to investigate the localization of Majorana modes, we rely on another technique: probing the chain with an additional quantum dot[24,38,39]. This can lift the ground-state degeneracy when coupled to two MBSs, while the system remains degenerate if it couples to one Majorana only.

### Assessing Majorana localization

We form an additional quantum dot on the left of the device (shown in red in the schematics of Fig. 4). Its energy levels are controlled by $V_{AD}$. When the QD levels are far from zero energy, the QD acts as a tunneling spectroscopy barrier[36]; instead, when a level is close to zero energy, it can couple to the Majoranas localized in the first dot of the chain, D1. Therefore, a simple QD-test involves sweeping $V_{AD}$ so that one QD energy level is brought on- and off-resonance. If the unpaired Majorana modes $\gamma_1$ and $\gamma_{2N}$ are perfectly localized at the chain ends, then the ZBP should not be perturbed. If, instead, there is a finite Majorana overlap

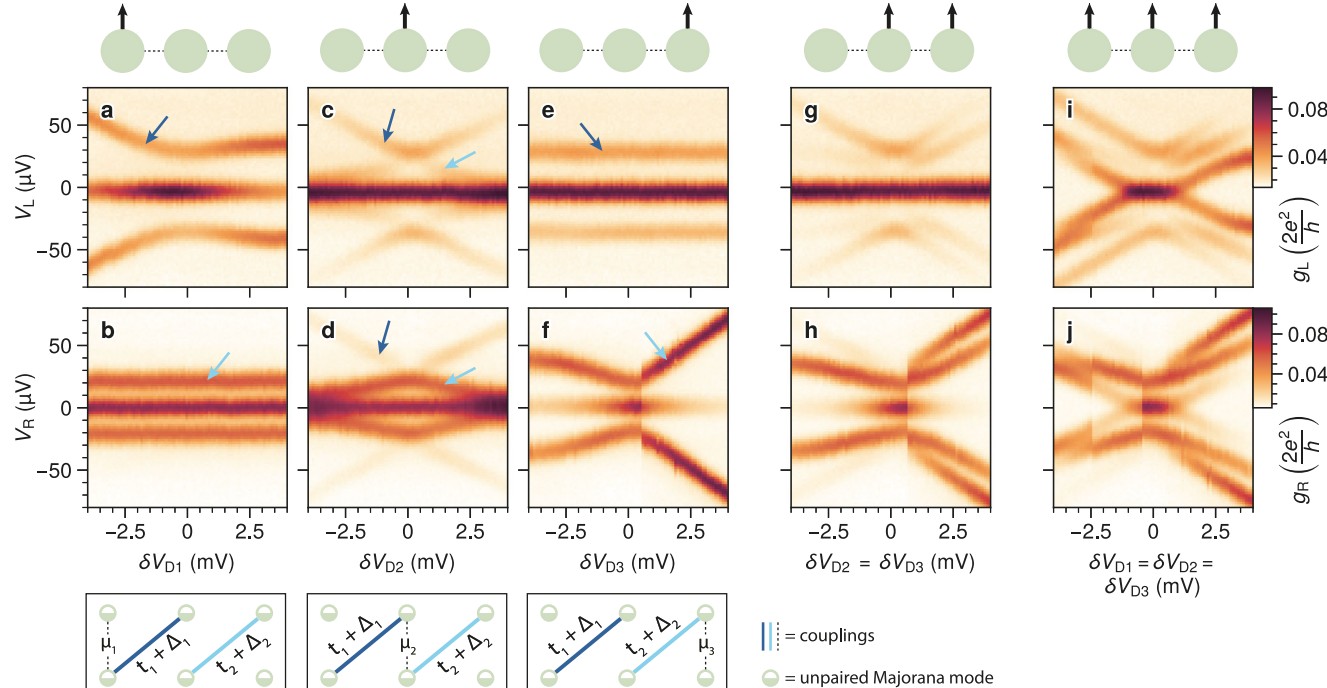

**Fig. 3 | Characterization of three-site chain spectra as a function of various QD detunings. a**–**f** Conductance spectroscopy of the three-site chain, measured from the left and the right, as a function of each QD plunger gate. The schematics below illustrate the corresponding Kitaev chain model in the Majorana basis. Color-coded arrows indicate the excited states illustrated in the schematics. **g**, **h** Left and right spectra as D2 and D3 are detuned simultaneously. **i**, **j** Left and right spectra while detuning all QDs of the chain.

in D1 then the Majoranas can couple through the additional dot and, therefore, the ZBP should broaden or even split.

We first look at the scenario where the chains are detuned on purpose (left side of Fig. 4), to make sure that we are able to resolve a ZBP splitting. Starting from the two-site chain, we observe in Fig. 4a, b a clear splitting of the lowest energy state. As predicted in ref. 24, this comes in two flavors known as "bow-tie" and "diamond"[38]. If both D1 and D2 are detuned (Fig. 4a), then the ZBP is already split when AD is off-resonance, and there is a zero-energy crossing as AD passes through a resonance (bow-tie shape). If only D2 is detuned (Fig. 4b), there is no splitting when AD is off-resonance and a finite splitting when AD is on-resonance (diamond shape). The diamond case is particularly insightful, as the ZBP wouldn't split without the additional quantum dot[4]: if a single QD is detuned from a poor man's Majorana sweet spot, the ZBP revealed by a standard tunneling spectroscopy persists even though the MBSs overlap in one dot[12,17]. This stresses the strength of this QD-test: it can reveal a local Majorana overlap even where standard tunneling spectroscopy fails to detect it.

On the other hand, the right side of Fig. 4 shows the situation where our Kitaev chain is tuned to the sweet spot: here, the QD-test does not resolve any ZBP splitting within its linewidth, indicating a strong localization of the Majorana modes. Theoretical simulations at the sweet spot replicate the spectral dependence. Furthermore, they can be used to extract microscopic parameters such as the coupling between AD and D1. As discussed in Methods, we fit the spinless model of Eq. (1) as well as a larger, spinful model including a second spin species at higher energies[28,44]. Both models qualitatively reproduce the $V_{AD}$ dependence. The values of the fitted parameters are reported in the Supplementary Information (Tables I and II). In particular, we extract a strong coupling between AD and D1, with tunnel amplitudes of order ~50 μeV and a non-local charging energy $U_{nl}$ ~20 μeV.

Finally, we turn our attention to the three-site chain case (Fig. 4e–h). With a global detuning of all QDs (Fig. 4e), we retrieve ZBP

splitting similar to the two-site case. However, if only one QD is detuned (Fig. 4f), there is no splitting anymore; instead, a ZBP persists over the full $V_{AD}$ range. We note that this is not a special property of D2: a ZBP persists if any of the three-site chain dots is detuned (Fig. S3). In this sense, the three-site chain is more resilient than two-site chains: Majoranas with a high degree of localization persist even if one of the QDs is off-resonance.

When all QDs are on resonance (Fig. 4g), the spectrum looks very similar to the two-site case. In particular, it seems that next-nearest-neighbor couplings are not able to split the ZBP in Fig. 4g, as far as this QD-test can resolve, even when looking carefully at the variations of the ZBP linewidth as $V_{AD}$ is varied (Fig. S16). We do not observe any ZBP broadening in any of the three-site chain sweet spots we tested: the ZBP half-width at half-maximum is 7 ± 1 μV, constantly (Fig. S16).

All the QD-test phenomenology shown here in Fig. 4 is reproduced in another Kitaev chain sweet spot within the same device, having different quantum dot orbitals and hybrid gate settings. The corresponding characterization and QD-tests are reported in Supplementary Information (Figs. S11 to S15).

## Discussion

### Comparison to long nanowires
A popular strategy for Majorana research involves continuous hybrid nanowires[45,46] rather than QD-based Kitaev chains. However, the material disorder in long nanowires complicates the unambiguous identification of MBSs in such systems[47–50]. In contrast, the site-by-site tunability of QD-based Kitaev chains can compensate for material inhomogeneities[51,52]. This leads to discrete and localized excitation spectra (Fig. 3), where every state can be interpreted with simple models[1,2,11,28].

With this tunability, we can even simulate disorder - in the form of deliberate perturbations to the system - and study its impact on the spectrum (Figs. 3 and 4). In short chains, such perturbations can split

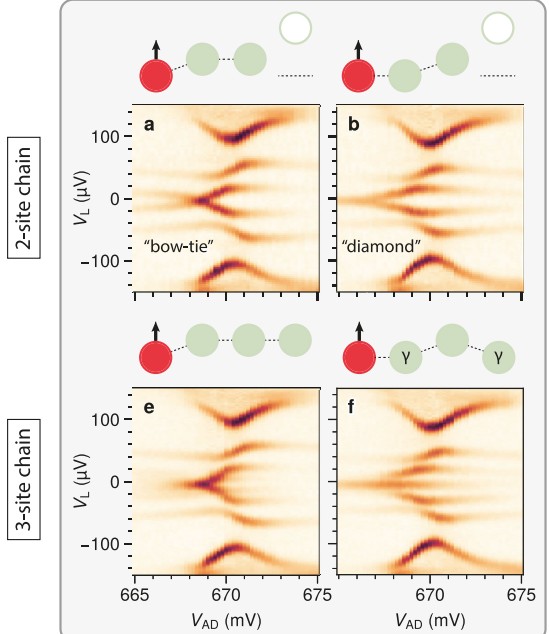

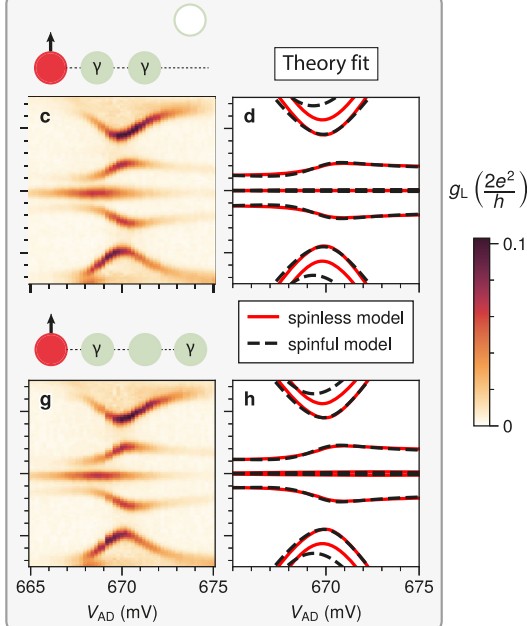

**Fig. 4 | Assessing Majorana localization in two- and three-site chains (first and second row, respectively).** Here, the Kitaev chains are coupled to an additional quantum dot, shown in red in the schematics. On the right side, the chains are tuned to the sweet spot. On the left side, they are detuned on purpose in various combinations of $\mu_n \neq 0$. This simple QD test - measuring the spectrum as a function of the gate voltage $V_{AD}$ controlling the additional dot - can expose delocalized Majoranas: if there is a finite Majorana overlap, the coupling to the additional QD splits the ZBP (panels **a**, **b**, and **e**), otherwise, highly localized Majoranas result in persistent ZBPs over the full $V_{AD}$ range (panels **c**, **f**, **g**). Panels (**d**) and (**h**) report a fit of panels (**c**) and (**g**) data, respectively, with the spinless model of Eq. (1) as well as a spinful model discussed in Methods and in the Supplementary Information. Theoretical simulations for all panels are reported in Figs. S2 and S3.

the ground-state degeneracy. However, if the system is perfectly tuned to the sweet spot, the Majoranas are localized in the outer QDs; thus, their wavefunctions do not overlap[11]. This is a fundamental difference compared to continuous nanowires, where the Majorana wavefunctions decay over a characteristic length scale $\xi$. This could lead to a detrimental Majorana overlap even in μm-long nanowires[53].

### Limitations of our device
In Figs. 2 and 3, we note that the energy gap isn't as large as in other devices[17], especially on the right side ($|2t_2| = |2\Delta_2| \approx 20 \,\mu eV$). This is limiting the impact of possible next-nearest-neighbor couplings, which scale as $\sim \mathcal{O}(t_n/E_Z)^{13,54}$, but it makes the system more vulnerable to thermal excitations $\sim e^{-t_n/k_B T}$. Finding the best compromise between suppressing next-nearest-neighbor couplings and avoiding thermal excitations is an open question[54].

Regarding the phase dependence, we note finite deviations from 0 or $\pi$ phase at $B_z = 0$ with a standard deviation $\approx 0.5 \,mT/T_\varphi \approx 0.13\pi$. This does not affect three-site chains, but in long chains it can lead to a $\approx 50\%$ reduction of the topological gap[32]. This can be prevented by stricter sweet-spot selection to have $\varphi_n$ even closer to zero or by better understanding the causes of the deviations. For instance, they could be due to $B_x$ not being perfectly aligned or the system not being perfectly one-dimensional. We also note anomalous phase dependences of two-site chains (see Fig. S17).

Regarding the QD-test, we note the lack of a direct measurement of the AD spin polarization. In Figs. S16, S14, and S15, the spin is inferred from the theory model. Finally, we note the resolution limit of the QD-test, given by tunneling spectroscopy techniques in state-of-the-art dilution refrigerators. Considering the variations of the ZBP

width as an estimate of the splitting (Fig. S16), our resolution is limited to $\sim 1\,\mu V$. Possible ways to go beyond such a resolution include the implementation of cQED spectroscopy techniques or, directly, the creation of a parity qubit made of two coupled Kitaev chains[4,28,30].

### Future directions
In this manuscript, we showed examples of $\varphi \approx 0$ and $\varphi \approx \pi$ at $B_z = 0$, but we haven't yet attempted to switch this behavior deterministically[32]. For instance, this could be implemented by switching the QD spins[43].

Regarding the QD-test, it would be illuminating to study the energy splitting as a function of the Zeeman energy. Next-nearest-neighbor hoppings should become relevant in the $E_Z - t_n$, $\Delta_n$ regime.

On the theory side, we limited the discussion to the qualitative reproduction of the spectra, well captured by the spinless Kitaev model of Eq. (1), and the extraction of the Hamiltonian parameters reported in Tables I and II (Supplementary Information). In future studies, it can be insightful to quantify the Majorana quality in terms of the Majorana polarization[23] or other quality measures[9]. This is particularly important with respect to braiding proposals, which have strict Majorana polarization requirements[28].

### Conclusion
In summary, we realized two- and three-site Kitaev chains within the same nanowire device. The relative phase $\varphi$ is tuned by the out-of-plane magnetic field $B_z$. We also show that it is possible to have either $\varphi \approx 0$ or $\varphi \approx \pi$ at $B_z = 0$. This demonstrates the possibility of controlling the phase of arbitrarily long Kitaev chains with appropriate sweet-spot choices, eliminating the need for cumbersome flux control[32]. At $\varphi = 0$,

we characterized the spectra of three-site chains under local and global perturbation of the quantum dots, finding unprecedented correspondence between the experiment and the Kitaev model, for the ground state and all the excited states.

Finally, we investigated the robustness of Majorana bound states formed at the chain ends against the simplest external perturbation of the Kitaev Hamiltonian: one extra energy level provided by an additional quantum dot. This QD-test is sensitive to the overlap of the two unpaired Majoranas on one side of the device, even when standard tunneling spectroscopy fails to detect it. Potential overlap of Majorana wavefunctions can cause the ZBP to split, which we do not observe unless the quantum dots of our Kitaev chains are deliberately detuned.

## Methods

### Nanofabrication
The device presented in this work is fabricated using the shadow-wall lithography technique[55,56]. The nanowire was placed onto a pre-patterned substrate, with Ti/Pd gates covered with an ALD-grown dielectric (20 nm $Al_2O_3$ and 10 nm $HfO_2$). A 17.5 nm layer of Al was deposited on the nanowire at a 30-degree angle with respect to the substrate, followed by a controlled in-situ oxidation at an $O_2$ over-pressure of 210 mTorr for 5 minutes. A more detailed description of the Al deposition can be found in ref. 57. Transene D was used to selectively remove the Al layer outside the nanowire region, which was protected by a PMM layer during the etching. Finally, the Cr/Au ohmic contacts were deposited following Ar milling.

### Theoretical modeling
We model the system using an extended spinless Kitaev chain coupled to an additional quantum dot. The total Hamiltonian is written as

$$H = H_N + H_{AD} + H_{tunnel}, \qquad (2)$$

$$H_{AD} = (\mu_0 - \mu_{offset})c_0^\dagger c_0, \qquad (3)$$

$$H_{tunnel} = t_d c_0^\dagger c_1 + \Delta_d c_0^\dagger c_1^\dagger + U_{nl} n_0 n_1 + \text{h.c.}, \qquad (4)$$

where $H_N$ describes the Kitaev chain. The additional dot is represented by a single fermionic level $c_0$ with chemical potential $\mu_0$, which is controlled experimentally via the gate voltage $V_{AD}$ as $(\mu_0 - \mu_{offset}) = -e\alpha_0 \delta V_{AD}$ where $\alpha_0$ is the lever arm. A small offset $\mu_{offset}$ is included to account for imperfect centering of the experimental gate-voltage window. The dot is coupled to the first site of the Kitaev chain through both normal tunneling $t_d$ and an effective superconducting pairing $\Delta_d$. In addition, we include a non-local Coulomb interaction, $U_{nl}$, between the dot and the adjacent chain site.

To assess the robustness of the spinless description, we also study a spinful extension that includes Zeeman splitting, local charging energy, and spin-orbit coupling, which introduce additional sources of Majorana hybridization. The model incorporates spin-dependent hopping and pairing terms parameterized by a spin-orbit angle $\theta$, as well as onsite and non-local interactions. For fixed Zeeman energy and charging energy, the system must be numerically tuned to a sweet spot that depends on $\theta$. Details of the Hamiltonian, parameter choices, and tuning procedure are provided in the Supplementary Information.

### Fit of experimental data
To fit the experimental differential-conductance spectra, we extract peak positions as a function of the additional-dot gate voltage using a numerical peak-finding algorithm and retain only continuous dispersing features. For a given set of microscopic parameters, we compute the many-body excitation spectrum of the model and match it to the extracted peaks, keeping only transitions with finite weight on the additional dot. The optimal parameters are obtained by minimizing the squared deviation between experimental and theoretical excitation energies using a differential evolution optimizer. To avoid overfitting, most chain parameters are fixed based on the experimental configuration, and we fit only a small subset of unknown parameters. See Supplementary Information for additional details and tables of the optimized parameter values.

### Extraction of the zero-field phase shift
In Fig. 2e–g we extract the distribution of the phase shift $\varphi_0$ at zero out-of-plane magnetic field $B_z = 0$. To this end, we consider a three-site Kitaev chain described by Eq. (1), tuned exactly to the sweet spot with $\mu_i = 0$, $t_1 = |\Delta_1|$, and $t_2 = |\Delta_2|e^{i(\varphi - \varphi_0)}$. In this limit, the Hamiltonian can be diagonalized analytically, yielding two non-trivial excitation energies due to particle-hole symmetry:

$$E_{\pm,\varphi}(\Delta_1, \Delta_2, \varphi_0) =$$
$$\sqrt{2(\Delta_1^2 + \Delta_2^2) \pm 2\sqrt{\Delta_1^4 - 2\Delta_1^2\Delta_2^2\cos(\varphi - \varphi_0) + \Delta_2^4}}.$$

We extract $\varphi_0$ by fitting these analytical expressions to the two lowest excited states obtained from the conductance spectra, minimizing a least-squares cost function. The phase $\varphi$ is inferred from the applied out-of-plane field. Details of the spectral post-processing and full fitting results are reported in the Supplementary Information.

## Data availability
All the data used in this study are available in the Zenodo database under accession code https://doi.org/10.5281/zenodo.15168550. This includes all the transport measurements discussed in the main text and Supplementary Information, as well as all other transport measurements ever recorded on the reported device.

## Code availability
The analysis code used to generate all figures of the publication is available at https://doi.org/10.5281/zenodo.15168550.

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

## Acknowledgements

This work has been supported by the Dutch Organization for Scientific Research (NWO) and Microsoft Corporation Station Q. R.S.S. acknowledges the Spanish Comunidad de Madrid (CM) "Talento Program" (Project No. 2022-T1/IND-24070), and grants No. PID2022-140552NA-I00 and No. CEX2024-001445-S funded by MICIU/AEI/10.13039/501100011033. We acknowledge Michael Wimmer for fruitful discussions and Sasa Gazibegovic for contributions to the nanowire growth.

## Author contributions

B.R. and N.v.L. fabricated the device with help from A.B. and F.Z. F.J.B.E. performed the transport measurements with help from A.B., F.Z., B.R., and N.v.L. J.D.T.L. performed the numerical simulations with help from R.S.S. and C.X.L. G.B. and E.P.A.M.B. provided the nanowires. W.D.H. and S.L.D.t.H. helped understand the phase behavior. A.B., G.P.M., N.v.L., and R.S.S. initiated the project. L.P.K. supervised the project. A.B. and F.JB.E. prepared the manuscript with help from B.R., J.D.T.L., N.v.L., L.P.K. and inputs from all authors.

## Competing interests

The authors declare no competing interests.
