## [Transparent Peer Review file · Nature Communications]

Probing Majorana localization of a phase-controlled three-site Kitaev chain with an additional quantum dot

Corresponding Author: Dr Alberto Bordin

Version 0:

Reviewer comments:

Reviewer #2

(Remarks to the Author)
In attachment

Reviewer Assessment - NCOMMS-25-78174-T

I've carefully read the authors response and modifications. The authors have answered satisfactorily to most of the points and issues that I have raised. They modified accordingly their manuscript, which results in an improved article. On the other hand, I believe this work is well adapted to Nature Communications in term of novelty and impact. Therefore, **I recommend its publication, granted that the two modifications asked below are performed.** Please find below my point-by-point response (in red) to the authors.

Comments / answers:

1.a. The procedure to extract the 0 and π phases is explained in Fig. ED13. It says that there are "identified manually by comparing the measured spectra with simulations". This manual procedure seems a bit vague, especially as the theoretical simulations are not shown. Can the authors change this procedure by defining a quantitative criterion to extract $\phi(B_z=0)$? More specifically, in some panels (h, c/i, m, u), it seems very difficult to extract something convincing. Can the authors comment on that?

We agree with the referee that in some instances it is difficult to extract the zero and π phases, in particular for sweet-spots 3 and 7. To improve our extraction procedure, we follow the referee's suggestion and introduce an automatic extraction algorithm. This is described in the newly added Supplementary Information section "Fit of the phase dependence of the three site Kitaev chain" and shown in the newly added figures ED3 and ED4. The automatically extracted phases are consistent with the values extracted manually.

Did the authors use this new automated extraction procedure in Fig. 2g? It does not look changed at all, i.e. still with the manual extraction procedure. If not, can the authors update Fig. 2g using the automatic extraction algorithm?

Indeed, we had not changed Fig. 2g yet, we have now updated it as the referee requested, using the automatic extraction algorithm.

1.b. The takeaway message of Fig. 2g given in the main text is that a binary nature is observed for the phase at $B_z=0$. I find this conclusion too strong as Fig. 2g shows that ϕ takes on different values depending on the sweet spot considered, some values going up to $1/4$ of $T\phi/2$, i.e. far from $\phi=0$. Moreover, for such a discussion it would be better to plot the extracted $\phi(B_z=0)$ vs sweetspot. Can the author make such a plot, including an error bar on $\phi(B_z=0)$ associated with the extraction procedure?

The most important result we want to highlight in the main text is not a precisely binary nature but rather the possibility of a full-electrical control of the phase. We note that this would be achievable even if the phases would be completely random: Liu et al. (2025, PRR) showed that as long as the phase differences are $\leq \pi/2$, then there is a topological gap in the long-chain limit. A perfect binary nature is desirable, but not necessary from a technological point of view.

In accordance, we softened our wording from "binary" to "bimodal" to reflect that our phases are not perfectly 0 or π .

Nevertheless, it seems that they are not completely random, as the automatic extraction of $\phi(B_z=0)$ of Fig. ED4 confirms, and this is interesting from a physical point of view, as we elaborate on below.

Ok

1.c. Finally, can the authors comment on the origin of this variation of the phase at zero perpendicular field? Vortex? In-plane magnetic field? π -junction physics? Role of the QD spin as mentioned in the conclusion?

As anticipated by Liu et al. (2025, PRR), the $0/\pi$ values for the superconducting phase difference in a three-site Kitaev chain are protected by complex conjugate symmetry. This symmetry is present in a

quasi-one-dimensional nanowire with spin-orbit interaction in proximity with conventional s-wave superconductivity, as discussed in ref. [Tewari and Sau 2012]. Here, in a practical semi-super device, the reported deviations from 0 and π indicate the breaking of such a symmetry. The possible physical mechanisms making the Hamiltonian complex include the direction of the applied magnetic field not being precisely perpendicular to the spin-orbit field, or the magnetic orbital effects on the wave functions of the QD and ABS. In the revised manuscript, we now mention these possibilities. Ongoing work on a 2DEG sample will investigate such mechanisms in further detail [Huisman et al. 2025, in preparation].

I'm happy with these modifications.

2. Fig. 4 compares the case of a detuned Kitaev chain (left panel) and a Kitaev chain at the sweet spot (right panel), by looking at the ZBP while AD is tuned at resonance with D1:

- Fig. 4c (2 sites, sweet spot) shows an absence of energy splitting of the ZBP while Fig. 4b (2 sites, detuned) shows a splitting. This supports the idea that the effect of a local external perturbation (AD1) strongly differs depending on the overlap of the MBS wavefunctions.
- Fig. 4g (3 sites, sweet spot) shows again an absence of energy splitting of the ZBP. However, Fig. 4f (3 sites, detuned) behaves differently as Fig. 4b discussed above as no energy splitting of the ZBP is observed. Since in that case D2 is detuned, the fermionic state in D1 should be trivial and sensitive to a coupling to AD, i.e. exhibiting an energy splitting, as in the 2-site case. This 2nd comparison, in the case of the 3-site Kitaev chain, does not seem to lead to the same conclusion. Can the authors comment on that? In particular, can the authors provide a theoretical simulation, similar to Fig. 4h, but in the case of the detuned Kitaev chain of Fig. 4f?

The referee is correctly pointing out that a large D2 detuning would interrupt the chain in the middle: it would break the three-site chain into two trivial fermionic states, one in D1 and one in D3. For smaller D2 detunings, however, a finite gap remains between the ZBP and the excited states, and the chain should still be considered as a whole. The transition can be appreciated in Fig. 3c,d, where the excitation gap closes for $\delta VD2 \sim 4$ mV. In Fig. 4f, VD2 was detuned only by 2 mV.

Therefore, the spectrum of Fig. 4f is not surprising and is in line with what predicted in ref. [Dourado et al. 2025]: if the middle QD of a three-site chain is slightly detuned, the Majorana sweet-spot persists. The Majorana polarization can even increase with increasing D2 detuning as the three-site chain is gradually transformed into an effective two-site chain using D2 as a virtual mediator of the D1-D3 couplings [Dourado et al. 2025]. We provide a theoretical simulation of this scenario in Supplementary Figure ED2k.

To improve the manuscript clarity, we now more clearly refer to the theoretical simulations in Figure ED2 and cite Dourado et al.

What the authors are explaining is quite subtle.

- **In place of the current Fig. 4f, can they provide a spectrum where D2 is further detuned that shows a splitting of the ZBP?** This would support better the message you are trying to convey.
- **If the authors cannot provide such a spectrum (because it has not been measured during the experimental run), can the authors be more explicit in the main text when discussing Fig. 4f? That one should expect a splitting, but that it is not observed here because the detuning VD2 is too small.** Currently, they only say *“However, if only one QD is detuned (Fig. 4f), there is no splitting anymore, instead, a ZBP persists over the full VAD range”, and “In this sense, the three-site chain is more resilient than two-site chains”.*

We cannot provide such a spectrum because, indeed, it has not been measured during the experimental run. So we updated the text discussing Fig. 4f to be more explicit as the referee suggested.

To make it even more clear, we added Fig. S2 and S3 in the Supplementary Information, reporting comparable detunings of VD1 and VD3, as well as a simulation of progressive VD2 detunings in Fig. S4.

Reviewer Assessment - NCOMMS-25-78174-T

I've carefully read the authors response and modifications. The authors have answered satisfactorily to most of the points and issues that I have raised. They modified accordingly their manuscript, which results in an improved article. On the other hand, I believe this work is well adapted to Nature Communications in term of novelty and impact. Therefore, **I recommend its publication, granted that the two modifications asked below are performed.** Please find below my point-by-point response (in red) to the authors.

Comments / answers:

1.a. The procedure to extract the 0 and π phases is explained in Fig. ED13. It says that there are "identified manually by comparing the measured spectra with simulations". This manual procedure seems a bit vague, especially as the theoretical simulations are not shown. Can the authors change this procedure by defining a quantitative criterion to extract $\phi(B_z=0)$? More specifically, in some panels (h, c/i, m, u), it seems very difficult to extract something convincing. Can the authors comment on that?

We agree with the referee that in some instances it is difficult to extract the zero and π phases, in particular for sweet-spots 3 and 7. To improve our extraction procedure, we follow the referee's suggestion and introduce an automatic extraction algorithm. This is described in the newly added Supplementary Information section "Fit of the phase dependence of the three site Kitaev chain" and shown in the newly added figures ED3 and ED4. The automatically extracted phases are consistent with the values extracted manually.

Did the authors use this new automated extraction procedure in Fig. 2g? It does not look changed at all, i.e. still with the manual extraction procedure. If not, can the authors update Fig. 2g using the automatic extraction algorithm?

1.b. The takeaway message of Fig. 2g given in the main text is that a binary nature is observed for the phase at $B_z=0$. I find this conclusion too strong as Fig. 2g shows that ϕ takes on different values depending on the sweet spot considered, some values going up to $1/4$ of $T\phi/2$, i.e. far from $\phi=0$. Moreover, for such a discussion it would be better to plot the extracted $\phi(B_z=0)$ vs sweetspot. Can the author make such a plot, including an error bar on $\phi(B_z=0)$ associated with the extraction procedure?

The most important result we want to highlight in the main text is not a precisely binary nature but rather the possibility of a full-electrical control of the phase. We note that this would be achievable even if the phases would be completely random: Liu et al. (2025, PRR) showed that as long as the phase differences are $\leq \pi/2$, then there is a topological gap in the long-chain limit. A perfect binary nature is desirable, but not necessary from a technological point of view.

In accordance, we softened our wording from "binary" to "bimodal" to reflect that our phases are not perfectly 0 or π .

Nevertheless, it seems that they are not completely random, as the automatic extraction of $\phi(B_z=0)$ of Fig. ED4 confirms, and this is interesting from a physical point of view, as we elaborate on below.

Ok

1.c. Finally, can the authors comment on the origin of this variation of the phase at zero perpendicular field? Vortex? In-plane magnetic field? π -junction physics? Role of the QD spin as mentioned in the conclusion?

As anticipated by Liu et al. (2025, PRR), the 0/ π values for the superconducting phase difference in a three-site Kitaev chain are protected by complex conjugate symmetry. This symmetry is present in a quasi-one-dimensional nanowire with spin-orbit interaction in proximity with conventional s-wave superconductivity, as discussed in ref. [Tewari and Sau 2012]. Here, in a practical semi-super device,

the reported deviations from 0 and π indicate the breaking of such a symmetry. The possible physical mechanisms making the Hamiltonian complex include the direction of the applied magnetic field not being precisely perpendicular to the spin-orbit field, or the magnetic orbital effects on the wave functions of the QD and ABS. In the revised manuscript, we now mention these possibilities. Ongoing work on a 2DEG sample will investigate such mechanisms in further detail [Huisman et al. 2025, in preparation].

I'm happy with these modifications.

2. Fig. 4 compares the case of a detuned Kitaev chain (left panel) and a Kitaev chain at the sweet spot (right panel), by looking at the ZBP while AD is tuned at resonance with D1:

- Fig. 4c (2 sites, sweet spot) shows an absence of energy splitting of the ZBP while Fig. 4b (2 sites, detuned) shows a splitting. This supports the idea that the effect of a local external perturbation (AD1) strongly differs depending on the overlap of the MBS wavefunctions.
- Fig. 4g (3 sites, sweet spot) shows again an absence of energy splitting of the ZBP. However, Fig. 4f (3 sites, detuned) behaves differently as Fig. 4b discussed above as no energy splitting of the ZBP is observed. Since in that case D2 is detuned, the fermionic state in D1 should be trivial and sensitive to a coupling to AD, i.e. exhibiting an energy splitting, as in the 2-site case. This 2nd comparison, in the case of the 3-site Kitaev chain, does not seem to lead to the same conclusion. Can the authors comment on that? In particular, can the authors provide a theoretical simulation, similar to Fig. 4h, but in the case of the detuned Kitaev chain of Fig. 4f?

The referee is correctly pointing out that a large D2 detuning would interrupt the chain in the middle: it would break the three-site chain into two trivial fermionic states, one in D1 and one in D3. For smaller D2 detunings, however, a finite gap remains between the ZBP and the excited states, and the chain should still be considered as a whole. The transition can be appreciated in Fig. 3c,d, where the excitation gap closes for $\delta VD2 \sim 4$ mV. In Fig. 4f, VD2 was detuned only by 2 mV.

Therefore, the spectrum of Fig. 4f is not surprising and is in line with what predicted in ref. [Dourado et al. 2025]: if the middle QD of a three-site chain is slightly detuned, the Majorana sweet-spot persists. The Majorana polarization can even increase with increasing D2 detuning as the three-site chain is gradually transformed into an effective two-site chain using D2 as a virtual mediator of the D1-D3 couplings [Dourado et al. 2025]. We provide a theoretical simulation of this scenario in Supplementary Figure ED2k.

To improve the manuscript clarity, we now more clearly refer to the theoretical simulations in Figure ED2 and cite Dourado et al.

What the authors are explaining is quite subtle.

- **In place of the current Fig. 4f, can they provide a spectrum where D2 is further detuned that shows a splitting of the ZBP?** This would support better the message you are trying to convey.
- **If the authors cannot provide such a spectrum (because it has not been measured during the experimental run), can the authors be more explicit in the main text when discussing Fig. 4f? That one should expect a splitting, but that it is not observed here because the detuning VD2 is too small.** Currently, they only say “*However, if only one QD is detuned (Fig. 4f), there is no splitting anymore, instead, a ZBP persists over the full VAD range*”, and “*In this sense, the three-site chain is more resilient than two-site chains*”.